# I Am Going MAD: Maximum Discrepancy Competition for Comparing Classifiers Adaptively

**Haotao Wang**
Department of Computer Science and Engineering
Texas A&M University
`htwang@tamu.edu`

**Tianlong Chen**
Department of Computer Science and Engineering
Texas A&M University
`wiwjp619@tamu.edu`

**Zhangyang Wang**
Department of Computer Science and Engineering
Texas A&M University
`atlaswang@tamu.edu`

**Kede Ma**
Department of Computer Science
City University of Hong Kong
`kede.ma@cityu.edu.hk`

## Abstract

The learning of hierarchical representations for image classification has experienced an impressive series of successes due in part to the availability of large-scale labeled data for training. On the other hand, the trained classifiers have traditionally been evaluated on small and fixed sets of test images, which are deemed to be extremely sparsely distributed in the space of all natural images. It is thus questionable whether recent performance improvements on the excessively re-used test sets generalize to real-world natural images with much richer content variations. Inspired by efficient stimulus selection for testing perceptual models in psychophysical and physiological studies, we present an alternative framework for comparing image classifiers, which we name the *MAximum Discrepancy* (MAD) competition. Rather than comparing image classifiers using fixed test images, we adaptively sample a small test set from an arbitrarily large corpus of unlabeled images so as to maximize the discrepancies between the classifiers, measured by the distance over WordNet hierarchy. Human labeling on the resulting model-dependent image sets reveals the relative performance of the competing classifiers, and provides useful insights on potential ways to improve them. We report the MAD competition results of eleven ImageNet classifiers while noting that the framework is readily extensible and cost-effective to add future classifiers into the competition. Codes can be found at `https://github.com/TAMU-VITA/MAD`.

## 1 Introduction

Large-scale human-labeled image datasets such as ImageNet (Deng et al., 2009) have greatly contributed to the rapid progress of research in image classification. In recent years, considerable effort has been put into designing novel network architectures (He et al., 2016; Hu et al., 2018) and advanced optimization algorithms (Kingma & Ba, 2015) to improve the training of image classifiers based on deep neural networks (DNNs), while little attention has been paid to comprehensive and fair evaluation/comparison of their model performance. Conventional model evaluation methodology for image classification generally follows a three-step approach (Burnham & Anderson, 2003). First, pre-select a number of images from the space of all possible natural images (*i.e.*, natural image manifold) to form the test set. Second, collect the human label for each image in the test set to identify its ground-truth category. Third, rank the competing classifiers according to their goodness of fit (*e.g.*, accuracy) on the test set; the one with the best result is declared the winner.

A significant problem with this methodology is the apparent contradiction between the enormous size and high dimensionality of natural image manifold and the limited scale of affordable testing (*i.e.*, human labeling, or verifying predicted labels, which is expensive and time consuming). As a result, a typical "large-scale" test set for image classification allows for tens of thousands of natural

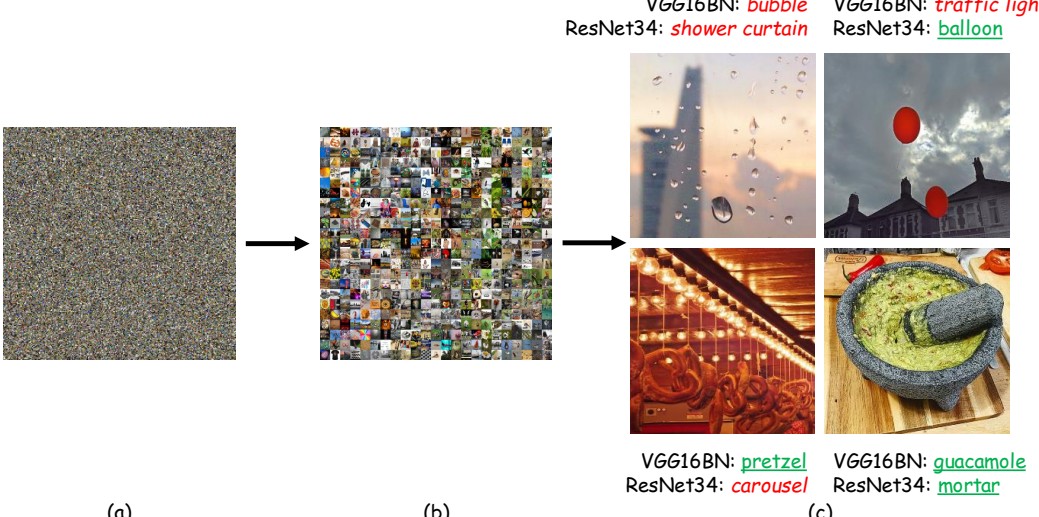

VGG16BN: *bubble*       VGG16BN: *traffic light*
ResNet34: *shower curtain*    ResNet34: balloon

VGG16BN: pretzel        VGG16BN: guacamole
ResNet34: *carousel*        ResNet34: mortar

(a)                (b)                (c)

Figure 1: Overview of the MAD competition procedure. **(a)**: A large unlabeled image set of web scale. **(b)**: The subset of natural images selected from (a) on which two classifiers (VGG16BN and ResNet34 in this case) make different predictions. Note that collecting the class label for each image in this subset may still be prohibitive because of its gigantic size. **(c)**: Representative examples sampled from top-$k$ images on which VGG16BN's and ResNet34's predictions differ the most, quantified by Eq. (3). Although the two classifiers have nearly identical accuracies on the ImageNet validation set, the proposed MAD competition successfully distinguishes them by finding their respective counterexamples. This sheds light on potential ways to improve the two classifiers or combine them into a better one. The model predictions are shown along with the images, where green underlined and *red italic* texts indicate correct and incorrect predictions, respectively.

images to be examined, which are deemed to be extremely sparsely distributed in natural image manifold. Model comparison based on a limited number of samples assumes that they are *sufficiently representative* of the whole population, an assumption that has been proven to be doubtful in image classification. Specifically, Recht et al. (2019) found that a minute natural distribution shift leads to a large drop in accuracy for a broad range of image classifiers on both CIFAR-10 (Krizhevsky, 2009) and ImageNet (Deng et al., 2009), suggesting that the current test sets may be far from sufficient to represent hard natural images encountered in the real world. Another problem with the conventional model comparison methodology is that the test sets are pre-selected and therefore fixed. This leaves the door open for adapting classifiers to the test images, deliberately or unintentionally, via extensive hyperparameter tuning, raising the risk of overfitting. As a result, it is never guaranteed that image classifiers with highly competitive performance on such small and fixed test sets can generalize to real-world natural images with much richer content variations.

In order to reliably measure the progress in image classification and to fairly test the generalizability of existing classifiers in a natural setting, we believe that it is necessary to compare the classifiers on a much larger image collection in the order of millions or even billions. Apparently, the main challenge here is how to exploit such a large-scale test set under the constraint of very limited budgets for human labeling, knowing that collecting ground-truth labels for all images is extremely difficult, if not impossible.

In this work, we propose an efficient and practical methodology, namely the *MAximum Discrepancy* (MAD) competition, to meet this challenge. Inspired by Wang & Simoncelli (2008) and Ma et al. (2019), instead of trying to *prove* an image classifier to be correct using a small and fixed test set, MAD starts with a large-scale unlabeled image set, and attempts to *falsify* a classifier by finding a set of images, whose predictions are in strong disagreement with the rest competing classifiers (see Figure 1). A classifier that is harder to be falsified in MAD is considered better. The initial image set for MAD to explore can be made arbitrarily large provided that the cost of computational prediction for all competing classifiers is cheap. To quantify the discrepancy between two classifiers on one image, we propose a weighted distance over WordNet hierarchy (Miller, 1998), which is more semantically aligned with human cognition compared with traditional binary judgment (agree

vs. disagree). The set of model-dependent images selected by MAD are the most informative in discriminating the competing classifiers. Subjective experiments on the MAD test set reveal the relative strengths and weaknesses among the classifiers, and identify the training techniques and architecture choices that improve the generalizability to natural image manifold. This suggests potential ways to improve a classifier or to combine aspects of multiple classifiers.

We apply the MAD competition to compare eleven ImageNet classifiers, and find that MAD verifies the relative improvements achieved by recent DNN-based methods, with a minimal subjective testing budget. MAD is readily extensible, allowing future classifiers to be added into the competition with little additional cost.

## 2 THE MAD COMPETITION METHODOLOGY

The general problem of model comparison in image classification may be formulated as follows. We work with the natural image manifold $\mathcal{X}$, upon which we define a class label $f(x) \in \mathcal{Y}$ for every $x \in \mathcal{X}$, where $\mathcal{Y} = \{1, 2, \ldots, c\}$ and $c$ is the number of categories. We assume a subjective assessment environment, in which a human subject can identify the category membership for any natural image $x$ among all possible categories. A group of image classifiers $\mathcal{F} = \{f_i\}_{i=1}^m$ are also assumed, each of which takes a natural image $x$ as input and makes a prediction of $f(x)$, collectively denoted by $\{f_i(x)\}_{i=1}^m$. The goal is to compare the relative performance of $m$ classifiers under very limited resource for subjective testing.

The conventional model comparison method for image classification first samples a natural image set $\mathcal{D} = \{x_k\}_{k=1}^n \subset \mathcal{X}$. For each image $x_k \in \mathcal{D}$, we ask human annotators to provide the ground-truth label $f(x_k) \in \mathcal{Y}$. Since human labeling is expensive and time consuming, and DNN-based classifiers are hungry for labeled data in the training stage (Krizhevsky et al., 2012), $n$ is typically small in the order of tens of thousands (Russakovsky et al., 2015). The predictions of the classifiers are compared against the human labels by computing the empirical classification accuracy

$$\text{Acc}(f_i; \mathcal{D}) = \frac{1}{|\mathcal{D}|} \sum_{x \in \mathcal{D}} \mathbb{I}[f_i(x) = f(x)], \text{ for } i = 1, \ldots, m. \tag{1}$$

The classifier $f_i$ with a higher classification accuracy is said to outperform $f_j$ with a lower accuracy.

As an alternative, the proposed MAD competition methodology aims to *falsify* a classifier in the most efficient way with the help of other competing classifiers. A classifier that is more likely to be falsified is considered worse.

### 2.1 THE MAD COMPETITION PROCEDURE

The MAD competition methodology starts by sampling an image set $\mathcal{D} = \{x_k\}_{k=1}^n$ from the natural image manifold $\mathcal{X}$. Since the number of images selected by MAD for subjective testing is independent of the size of $\mathcal{D}$, we may choose $n$ to be arbitrarily large such that $\mathcal{D}$ provides dense coverage of (*i.e.*, sufficiently represents) $\mathcal{X}$. MAD relies on a distance measure to quantify the degree of discrepancy between the predictions of any two classifiers. The most straightforward measure is the 0-1 loss:

$$d_{01}(f_i(x), f_j(x)) = \mathbb{I}[f_i(x) \neq f_j(x)]. \tag{2}$$

Unfortunately, it ignores the semantic relations between class labels, which may be crucial in distinguishing two classifiers, especially when they share similar design philosophies (*e.g.*, using DNNs as backbones) and are trained on the same image set (*e.g.*, ImageNet). For example, misclassifying a "chihuahua" as a dog of other species is clearly more acceptable compared with misclassifying it as a "watermelon". We propose to leverage the semantic hierarchy in WordNet (Miller, 1998) to measure the distance between two (predicted) class labels. Specifically, we model WordNet as a weighted undirected graph $G(V, E)$[1]. Each edge $e = (u, v) \in E$ connects a parent node $u$ of a more general level (*e.g.*, canine) to its child node $v$ of a more specific level (*e.g.*, dog), for $u, v \in V$. A nonnegative weight $w(e)$ is assigned to each edge $e = (u, v)$ to encode the semantic distance

---

[1]Although WordNet is tree-structured, a child node in WordNet may have multiple parent nodes.

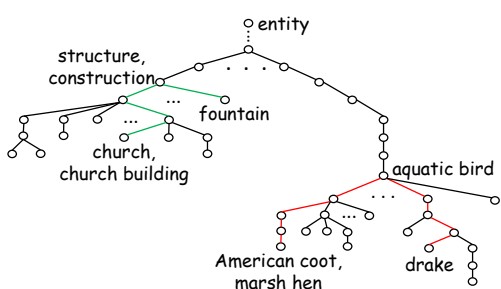

Figure 2: Comparison of weighted and un-weighted distances. In the sub-tree of Word-Net, we highlight the shortest paths from "fountain" to "church" and from "drake" to "American coot" in green and red, respectively. The semantic distance between the two aquatic birds is much shorter than that between the two constructions of completely different functionalities (verified by our internal subjective testing). The proposed weighted distance is well aligned with human cognition by assigning a much smaller distance to the red path (0.0037) compared with the green one (0.0859).

between $u$ and $v$. A larger $w(e)$ indicates that $u$ and $v$ are semantically more dissimilar. We measure the distance between two labels as the sum of the weights assigned to the edges along the shortest path $\mathcal{P}$ connecting them

$$d_w(f_i(x), f_j(x)) = \sum_{e \in \mathcal{P}} w(e). \tag{3}$$

Eq. (3) reduces to the standard graph hop distance between two vertices by setting $w(e) = 1$. We design $w(e)$ to be inversely proportional to the tree depth level $l$ of the parent node (*e.g.*, $w(e) \propto 2^{-l}$). In other words, we prefer the shortest paths to traverse the root node (or nodes with smaller $l$) as a way of encouraging $f_i(x)$ and $f_j(x)$ to differ in a more general level (*e.g.*, vehicle rather than watercraft). Figure 2 shows the semantic advantages of our choice of weighting compared with the equal weighting. With the distance measure at hand, the optimal image in terms of discriminating $f_i$ and $f_j$ can be obtained by maximizing the discrepancy between the two classifiers on $\mathcal{D}$

$$x^\star = \arg\max_{x \in \mathcal{D}} d_w(f_i(x), f_j(x)). \tag{4}$$

The queried image label $f(x^\star)$ leads to three possible outcomes (see Figure 1):

- **Case I**. Both classifiers make correct predictions. Although theoretically impossible based on the general problem formulation, it is not uncommon in practice that a natural image may contain multiple distinct objects (*e.g.*, guacamole and mortar). In this case, $f_i$ and $f_j$ successfully recognize different objects in $x^\star$, indicating that both classifiers tend to perform at a high level. By restricting $\mathcal{D}$ to only contain natural images with a single salient object, we may reduce the possibility of this outcome.

- **Case II**. $f_i$ (or $f_j$) makes correct prediction, while $f_j$ (or $f_i$) makes incorrect prediction. In this case, MAD automatically identifies a strong failure case to falsify one classifier, not the other; a clear winner is obtained. The selected image $x^\star$ provides the strongest evidence in differentiating the two classifiers as well as ranking their relative performance.

- **Case III**. Both classifiers make incorrect predictions in a multiclass image classification problem (*i.e.*, $c \geq 3$). Although both classifiers make mistakes, they differ substantially during inference, which in turn provides a strong indication of their respective weaknesses[2]. Depending on the subjective experimental setting, $x^\star$ may be used to rank the classifiers based on Eq. (3) if the full label $f(x^\star)$ has been collected. As will be clear later, due to the difficulty in subjective testing, we collect a partial label: "$x^\star$ does not contain $f_i(x^\star)$ nor $f_j(x^\star)$". In this case, only Eq. (1) can be applied, and $x^\star$ contributes less to performance comparison between the two classifiers.

---

[2]This is in stark contrast to natural adversarial examples in ImageNet-A (Hendrycks et al., 2019), where different image classifiers tend to make consistent mistakes. For example, VGG16BN and ResNet34 make the same incorrect predictions on $3,149$ out of $7,500$ images in ImageNet-A.

---

**Algorithm 1:** The MAD competition

---

**Input:** An unlabeled image set $\mathcal{D}$, a group of image classifiers $\mathcal{F} = \{f_i\}_{i=1}^m$ to be ranked, a distance measure $d_w$ defined over WordNet hierarchy

**Output:** A global ranking vector $r \in \mathbb{R}^m$

1   $\mathcal{S} \leftarrow \emptyset$, $B \leftarrow I$
2   **for** $i \leftarrow 1$ **to** $m$ **do**
3     Compute classifier predictions $\{f_i(x), x \in \mathcal{D}\}$
4   **end**
5   **for** $i \leftarrow 1$ **to** $m$ **do**
6     **for** $j \leftarrow i + 1$ **to** $m$ **do**
7       Compute the distances using Eq. (3) $\{d_w(f_i(x), f_j(x)), x \in \mathcal{D}\}$
8       Select top-$k$ images with $k$ largest distances to form $\mathcal{S}_{\{i,j\}}$
9       $\mathcal{S} \leftarrow \mathcal{S} \bigcup \mathcal{S}_{\{i,j\}}$
10     **end**
11   **end**
12   Source human labels for $\mathcal{S}$
13   Compute the pairwise accuracy matrix $A$ with $a_{ij} = \text{Acc}(f_i; \mathcal{S}_{\{i,j\}})$ using Eq. (1)
14   Compute the pairwise dominance matrix $B$ with $b_{ij} = a_{ij}/a_{ji}$
15   Compute the global ranking vector $r$ using Eq. (5)

---

**Algorithm 2:** Adding a new classifier into the MAD competition

---

**Input:** An unlabeled image set $\mathcal{D}$, the pairwise dominance matrix $B \in \mathbb{R}^{m \times m}$ for $\mathcal{F} = \{f_i\}_{i=1}^m$, a new classifier $f_{m+1}$ to be ranked, distance measure $d_w$

**Output:** A global ranking vector $r \in \mathbb{R}^{m+1}$

1   $\mathcal{S} \leftarrow \emptyset$,
    $B' \leftarrow \begin{bmatrix} B & 0 \\ 0^T & 1 \end{bmatrix} \in \mathbb{R}^{(m+1) \times (m+1)}$
2   Compute the predictions of the new image classifier $\{f_{m+1}(x), x \in \mathcal{D}\}$
3   **for** $i \leftarrow 1$ **to** $m$ **do**
4     Compute the distances using Eq. (3) $\{d_w(f_i(x), f_{m+1}(x)), x \in \mathcal{D}\}$
5     Select top-$k$ images with $k$ largest distances to form $\mathcal{S}_{\{i,m+1\}}$
6     $\mathcal{S} = \mathcal{S} \bigcup \mathcal{S}_{\{i,m+1\}}$
7   **end**
8   Source human labels for $\mathcal{S}$
9   **for** $i \leftarrow 1$ **to** $m$ **do**
10    $a_{i,m+1} = \text{Acc}(f_i; \mathcal{S}_{\{i,m+1\}})$
     $a_{m+1,i} = \text{Acc}(f_{m+1}; \mathcal{S}_{\{i,m+1\}})$
11   **end**
12   Update the pairwise dominance matrix $B'$ with $b'_{i,m+1} = 1/b'_{m+1,i} = a_{i,m+1}/a_{m+1,i}$
13   Compute the global ranking vector $r \in \mathbb{R}^{m+1}$ using Eq. (5)

---

In practice, to obtain reliable performance comparison between $f_i$ and $f_j$, we choose top-$k$ images in $\mathcal{D}$ with $k$ largest distances computed by Eq. (3) to form the test subset $\mathcal{S}_{\{i,j\}}$. MAD runs this game among all $\binom{m}{2}$ distinct pairs of classifiers, resulting in the final MAD test set $\mathcal{S} = \bigcup \mathcal{S}_{\{i,j\}}$. The number of natural images in $\mathcal{S}$ is at most $m(m-1)k/2$, which is independent of the size $n$ of $\mathcal{D}$. In other words, applying MAD to a larger image set has no impact on the cost of human labeling. In scenarios where the cost of computational prediction can be ignored, MAD encourages to expand $\mathcal{D}$ to cover as many "free" natural images as possible.

We now describe our subjective assessment environment for collecting human labels. Given an image $x \in \mathcal{S}$, which is associated with two classifiers $f_i$ and $f_j$, we pick two binary questions for human annotators: "Does $x$ contain an $f_i(x)$?" and "Does $x$ contain an $f_j(x)$?". When both answers are no (corresponding to Case III), we stop querying the ground-truth label of $x$ because it is difficult for humans to select one among $c$ classes, especially when $c$ is large and the ontology of classes is complex.

After subjective testing, we first compare the classifiers in pairs and aggregate the pairwise statistics into a global ranking. Specifically, we compute the empirical classification accuracies of $f_i$ and $f_j$ on $\mathcal{S}_{\{i,j\}}$ using Eq. (1), denoted by $a_{ij}$ and $a_{ji}$, respectively. When $k$ is small, Laplace smoothing is employed to smooth the estimation. Note that $a_{ij} + a_{ji}$ may be greater than one because of Case I. The pairwise accuracy statistics of all classifiers form a matrix $A$, from which we compute another matrix $B$ with $b_{ij} = a_{ij}/a_{ji}$, indicating the pairwise dominance of $f_i$ over $f_j$. We aggregate the pairwise comparison results into a global ranking $r \in \mathbb{R}^m$ using Perron rank (Saaty & Vargas, 1984):

$$r = \lim_{t \to \infty} \frac{1}{t} \sum_{\alpha=1}^{t} \frac{B^\alpha 1}{1^T B^\alpha 1}, \tag{5}$$

where $1$ is an $m$-dimensional vector of all ones. The limit of Eq. (5) is the normalized principal eigenvector of $B$ corresponding to the largest eigenvalue, where $r_i > 0$ for $i = 1, \ldots m$ and $\sum_i r_i = 1$. The larger $r_i$ is, the better $f_i$ performs in the MAD competition. Other ranking aggregation methods such as HodgeRank (Jiang et al., 2011) may also be applied. We summarize the workflow of the MAD competition in Algorithm 1.

Finally, it is straightforward and cost-effective to add the $m + 1$-th classifier into the current MAD competition. No change is needed for the sampled $\mathcal{S}$ and the associated subjective testing. The additional work is to select a total of $mk$ new images from $\mathcal{D}$ for human labeling. We then enlarge $B$ by one along its row and column, and insert the pairwise comparison statistics between $f_{m+1}$ and the previous $m$ classifiers. An updated global ranking vector $r \in \mathbb{R}^{m+1}$ can be computed using Eq. (5). We summarize the procedure of adding a new classifier in Algorithm 2.

## 3 APPLICATION TO IMAGENET CLASSIFIERS

In this section, we apply the proposed MAD competition methodology to comparing ImageNet classifiers. We focus on ImageNet (Deng et al., 2009) for two reasons. First, it is one of the first large-scale and widely used datasets in image classification. Second, the improvements on ImageNet seem to plateau, which provides an ideal platform for MAD to distinguish the newly proposed image classifiers finer.

### 3.1 EXPERIMENTAL SETUPS

**Constructing** $\mathcal{D}$   Inspired by Hendrycks et al. (2019), we focus on the same $200$ out of $1,000$ classes to avoid rare and abstract ones, and classes that have changed much since 2012. For each class, we crawl a large number of images from Flickr, resulting in a total of $n = 168,000$ natural images. Although MAD allows us to arbitrarily increase $n$ with essentially no cost, we choose the size of $\mathcal{D}$ to be approximately three times larger than the ImageNet validation set to provide a relatively easy environment for probing the generalizability of the classifiers. As will be clear in Section 3.2, the current setting of $n$ is sufficient to discriminate the competing classifiers. To guarantee the content independence between ImageNet and $\mathcal{D}$, we collect images that have been uploaded after 2013. It is worth noting that no data cleaning (*e.g.*, inappropriate content and near-duplicate removal) is necessary at this stage since we only need to ensure the selected subset $\mathcal{S}$ for human labeling is eligible.

**Competing Classifiers**   We select eleven representative ImageNet classifiers for benchmarking: VGG16BN (Simonyan & Zisserman, 2014) with batch normalization (Ioffe & Szegedy, 2015), ResNet34, ResNet101 (He et al., 2016), WRN101-2 (Zagoruyko & Komodakis, 2016), ResNeXt101-32×4 (Xie et al., 2017), SE-ResNet-101, SENet154 (Hu et al., 2018), NASNet-A-Large (Zoph et al., 2018), PNASNet-5-Large (Liu et al., 2018), EfficientNet-B7 (Tan & Le, 2019), and WSL-ResNeXt101-32×48 (Mahajan et al., 2018). Since VGG16BN and ResNet34 have nearly identical accuracies on the ImageNet validation set, it is particularly interesting to see which method generalizes better to natural image manifold. We compare ResNet34 with ResNet101 to see the influence of DNN depth. WRN101-2, ResNeXt101-32×4, SE-ResNet-101 are different improved versions over ResNet-101. We also include two state-of-the-art classifiers: WSL-ResNeXt101-32×48 and EfficientNet-B7. The former leverages the power of weakly supervised pre-training on Instagram data, while the latter makes use of compound scaling. We use publicly available code repositories for all DNN-based models, whose top-1 accuracies on the ImageNet validation set are listed in Table 1 for reference.

**Constructing** $\mathcal{S}$   When constructing $\mathcal{S}$ using the maximum discrepancy principle, we add another constraint based on prediction confidence. Specifically, a candidate image $x$ associated with $f_i$ and $f_j$ is filtered out if it does not satisfy $\min(p_i(x), p_j(x)) \geq T$, where $p_i(x)$ is the confidence score (*i.e.*, probability produced by the last softmax layer) of $f_i(x)$ and $T$ is a predefined threshold set to $0.8$. We include the confidence constraint for two main reasons. First, if $f_i$ misclassifies $x$ with low confidence, it is highly likely that $x$ is near the decision boundary and thus contains less information on improving the decision rules of $f_i$. Second, some images in $\mathcal{D}$ do not necessarily fall into the $1,000$ classes in ImageNet, which are bound to be misclassified (a problem closely related to out-of-distribution detection). If they are misclassified by $f_i$ with high confidence, we consider them as hard counterexamples of $f_i$. To encourage class diversity in $\mathcal{S}$, we retain a maximum of three images

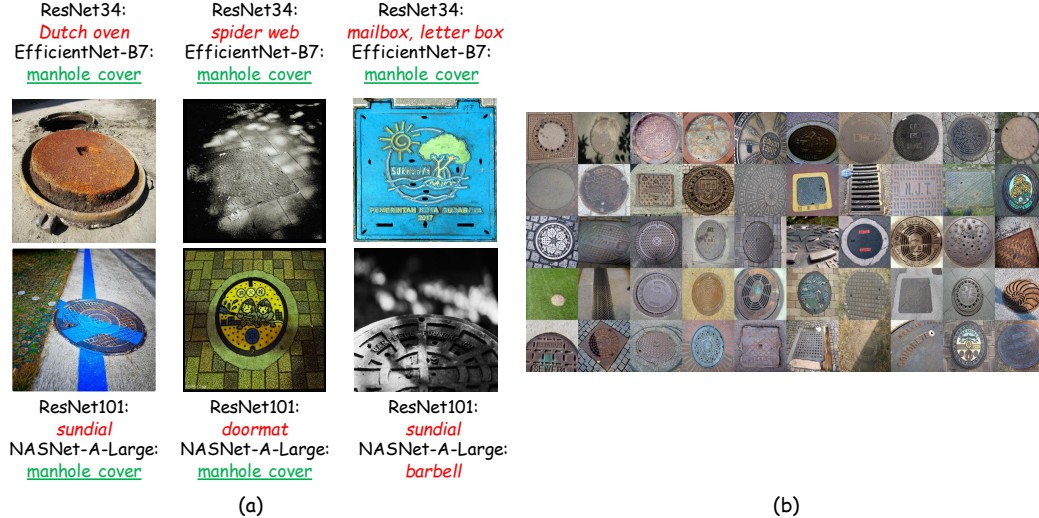

Figure 4: Visual comparison of images selected by MAD and in the ImageNet validation set. **(a)**: "manhole cover" images selected by MAD along with the predictions by the associated classifiers. **(b)**: All "manhole cover" images in the ImageNet validation set. The MAD-selected images are visually much harder, which contain diverse and non-trivial distortions, *e.g.*, occlusion, shading and unbalanced lightening, complex background, rare colors, and untypical viewing points. In contrast, ImageNet images mainly include a single center-positioned object with a relatively clean background, whose shape, color and viewing point are common.

with the same predicted label by $f_i$. In addition, we exclude images that are non-natural. Figure 4 visually compares representative "manhole cover" images in $\mathcal{S}$ and the ImageNet validation set (see more in Figure 6).

**Collecting Human Labels**  As described in Section 2.1, given an image $x \in \mathcal{S}$, human annotators need to answer two binary questions. In our subjective experiments, we choose $k = 30$ and invite five volunteer graduate students, who are experts in computer vision, to label a total of $11 \times 10 \times 30/2 = 1,650$ images. If more than three of them find difficulty in labeling $x$ (associated with $f_i$ and $f_j$), it is discarded and replaced by $x' \in \mathcal{D}$ with the $k + 1$-th largest distance $d_w(f_i(x'), f_j(x'))$. Majority vote is adopted to decide the final label when disagreement occurs. After subjective testing, we find that 53.5% of annotated images belong to Case II, which form the cornerstone of the subsequent data analysis. Besides, 32.9% and 13.6% images pertain to Case I and Case III, respectively.

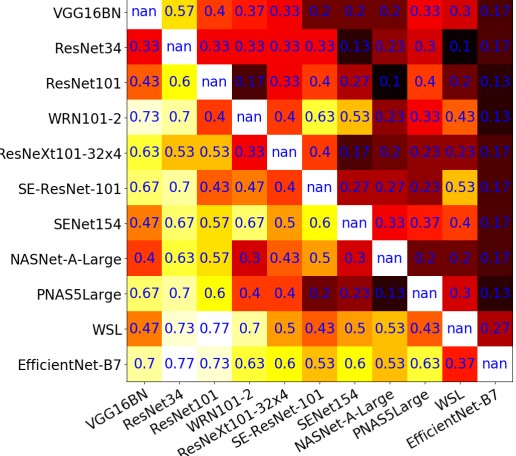

Figure 3: Pairwise accuracy matrix $A$ with brighter colors indicating higher accuracies (blue numbers). WSL-ResNeXt101-32×48 is abbreviated to WSL for neat presentation.

## 3.2 EXPERIMENTAL RESULTS

**Pairwise Ranking Results**  Figure 3 shows the pairwise accuracy matrix $A$ in the current MAD competition, where a larger value of an entry (a brighter color) indicates a higher accuracy in the subset selected together by the corresponding row and column models. An interesting phenomenon we observe is that when two classifiers $f_i$ and $f_j$ perform at a similar level on $\mathcal{S}_{\{i,j\}}$ (*i.e.*, $|\text{Acc}(f_i) - \text{Acc}(f_j)|$ is small), $\max(\text{Acc}(f_i), \text{Acc}(f_j))$ is also small. That is, more images on which they both make incorrect but different predictions (Case III) have been selected compared with images falling into Case I. Taking a closer look at images in $\mathcal{S}_{\{i,j\}}$, we may reveal the respective model biases of $f_i$

| Models | ImageNet top-1 Acc | Acc Rank | MAD Rank | △ Rank |
|---|---|---|---|---|
| WSL-ResNeXt101-32×48 (Mahajan et al., 2018) | 85.44 | 1 | 2 | -1 |
| EfficientNet-B7 (Tan & Le, 2019) | 84.48 | 2 | 1 | 1 |
| PNASNet-5-Large (Liu et al., 2018) | 82.74 | 3 | 7 | -4 |
| NASNet-A-Large (Zoph et al., 2018) | 82.51 | 4 | 4 | 0 |
| SENet154 (Hu et al., 2018) | 81.30 | 5 | 3 | 2 |
| WRN101-2 (Zagoruyko & Komodakis, 2016) | 78.85 | 6 | 6 | 0 |
| SE-ResNet-101 (Hu et al., 2018) | 78.40 | 7 | 5 | 2 |
| ResNeXt101-32×4 (Xie et al., 2017) | 78.19 | 8 | 8 | 0 |
| ResNet101 (He et al., 2016) | 77.37 | 9 | 10 | -1 |
| VGG16BN (Simonyan & Zisserman, 2014) | 73.36 | 10 | 9 | 1 |
| ResNet34 (He et al., 2016) | 73.31 | 11 | 11 | 0 |

Table 1: Global ranking results. A smaller rank indicates better performance.

and $f_j$. For example, we find that WSL-ResNeXt101-32×48 tends to focus on foreground objects, while EfficientNet-B7 attends more to background objects (see Figure 7). We also find several common failure modes of the competing classifiers through pairwise comparison, *e.g.*, excessive reliance on relation inference (see Figure 8), bias towards low-level visual features (see Figure 9), and difficulty in recognizing rare instantiations of objects (see Figures 4 and 6).

**Global Ranking Results** We present the global ranking results by MAD in Table 1, where we find that MAD tracks the steady progress in image classification, as verified by a reasonable Spearman rank-order correlation coefficient (SRCC) of 0.89 between the accuracy rank on the ImageNet validation set and the MAD rank on our test set $\mathcal{D}$. Moreover, by looking at the differences between the two rankings, we obtain a number of interesting findings. First, VGG16BN outperforms not only ResNet34 but also ResNet101, suggesting

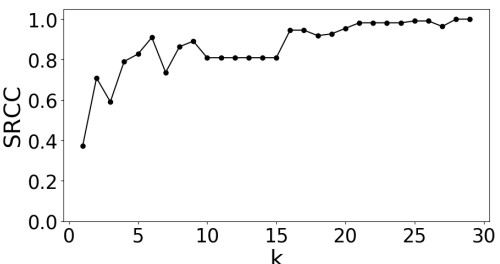

Figure 5: The SRCC values between top-30 and other top-$k$ rankings, $k = \{1, 2, \ldots, 29\}$.

that under similar computation budgets, VGG-like networks may exhibit better generalizability to hard samples than networks with residual connections. Second, both networks equipped with the squeeze-and-extraction mechanism, *i.e.*, SE-ResNet-101 and SENet154, move up by two places in the MAD ranking. This indicates that explicitly modeling dependencies between channel-wise feature maps seems quite beneficial to image classification. Third, for the two models that exploit neural architecture search, NASNet-A-Large is still ranked high by MAD; interestingly, the rank of PNASNet-5-Large drops a lot. This implies MAD may prefer the global search strategy used in NASNet-A-Large to the progressive cell-wise search strategy adopted in PNASNet-5-Large, although the former is slightly inferior in ImageNet top-1 accuracy. Last but not least, the top-2 performers, WSL-ResNeXt101-32×48 and EfficientNet-B7, are still the best in MAD competition (irrespective of their relative rankings), verifying the effectiveness of large-scale hashtag data pre-training and compound scaling in the context of image classification.

**Ablation Study** We analyze the key hyperparameter $k$ in MAD, *i.e.*, the number of images in $\mathcal{S}_{\{i,j\}}$ selected for subjective testing. We calculate the SRCC values between the top-30 ranking (as reference) and other top-$k$ rankings with $k = \{1, 2, \ldots, 29\}$. As shown in Figure 5, the ranking results are fairly stable (SRCC > 0.90) when $k > 15$. This supports our choice of $k = 30$ since the final global ranking already seems to enter a stable plateau.

## 4    DISCUSSION

We have presented a new methodology, MAD competition, for comparing image classification models. MAD effectively mitigates the conflict between the prohibitively large natural image manifold that we have to evaluate against and the expensive human labeling effort that we aim to minimize. Much of our endeavor has been dedicated to selecting natural images that are optimal in terms of distinguishing or falsifying classifiers. MAD requires explicit specification of image classifiers to

be compared, and provides an effective means of exposing the respective flaws of competing classifiers. It also directly contributes to model interpretability and helps us analyze the models' focus and bias when making predictions. We have demonstrated the effectiveness of MAD competition in comparing ImageNet classifiers, and concluded a number of interesting observations, which were not apparently drawn from the (often quite close) accuracy numbers on the ImageNet validation set.

The application scope of MAD is far beyond image classification. It can be applied to computational models that produce discrete-valued outputs, and is particularly useful when the sample space is large and the ground-truth label being predicted is expensive to measure. Examples include medical and hyperspectral image classification (Filipovych & Davatzikos, 2011; Wang et al., 2014), where signification domain expertise is crucial to obtain correct labels. MAD can also be used to spot rare but fatal failures in high-cost and failure-sensitive applications, *e.g.*, comparing perception systems of autonomous cars (Chen et al., 2015) in unconstrained real-world weathers, lighting conditions, and road scenes. In addition, by restricting the test set $\mathcal{D}$ to some domain of interest, MAD allows comparison of classifiers in more specific applications, *e.g.*, fine-grained image recognition.

We feel it important to note the limitations of the current MAD. First, MAD aims at relatively comparing models, and cannot give an absolute performance measure. Second, as an "error spotting" mechanism, MAD implicitly assumes that models in the competition are reasonably good (*e.g.*, ImageNet classifiers); otherwise, the selected counterexamples may be less meaningful. Third, although the distance in Eq. (3) is sufficient to distinguish multiple classifiers in the current experimental setting, it does not yet fully reflect human cognition of image label semantics. Fourth, the confidence computation used to select images is not perfectly grounded. How to marry the MAD competition with Bayesian probability theory to model uncertainties during image selection is an interesting direction for future research. Due to the above issues, MAD should be viewed as complementary to, rather than a replacement for, the conventional accuracy comparison for image classification.

Our method arises as a natural combination of concepts drawn from two separate lines of research. The first explores the idea of model falsification as model comparison. Wang & Simoncelli (2008) introduced the maximum differentiation competition for comparing computational models of *continuous* perceptual quantities, which was further extended by Ma et al. (2019). Berardino et al. (2017) developed a computational method for comparing hierarchical image representations in terms of their ability to explain perceptual sensitivity in humans. MAD, on the other hand, is tailored to applications with *discrete* model responses and relies on a semantic distance measure to compute model discrepancy. The second endeavour arises from machine learning literature on generating adversarial examples (Szegedy et al., 2013; Goodfellow et al., 2015; Madry et al., 2018) and evaluating image classifiers on new test sets (Geirhos et al., 2019; Recht et al., 2019; Hendrycks & Dietterich, 2019; Hendrycks et al., 2019). The images selected by MAD can be seen as a form of natural adversarial examples as each of them is able to fool at least one classifier (when Case I is eliminated). Unlike adversarial images with inherent transferability to mislead most classifiers, MAD-selected images emphasize on their discriminability of the competing models. Different from recently created test sets, the MAD-selected set is adapted to the competing classifiers with the goal of minimizing human labeling effort. In addition, MAD can also be linked to the popular technique of differential testing (McKeeman, 1998) in software engineering.

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

## A  MORE VISUALIZATION RESULTS

Due to the page limit, we put some additional figures here.

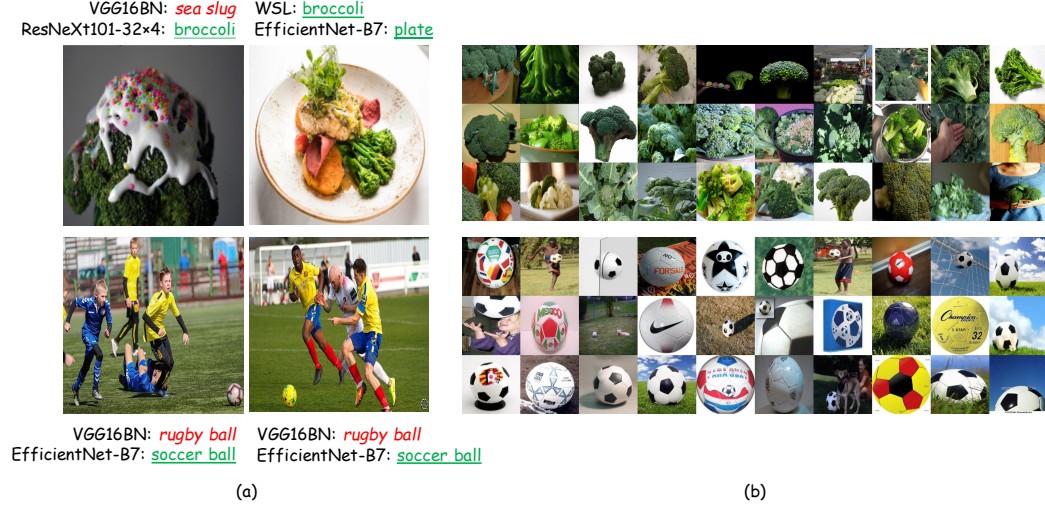

Figure 6: Visual comparison of "broccoli" and "soccer ball" images (a) selected by MAD and (b) in the ImageNet validation set.

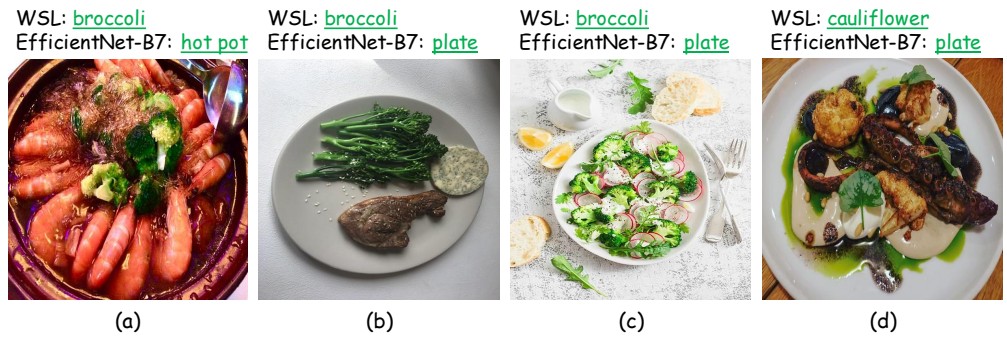

Figure 7: Examples of network bias. WSL-ResNeXt101-32×48 (WSL) tends to focus on foreground objects, while EfficientNet-B7 attends more to background objects.

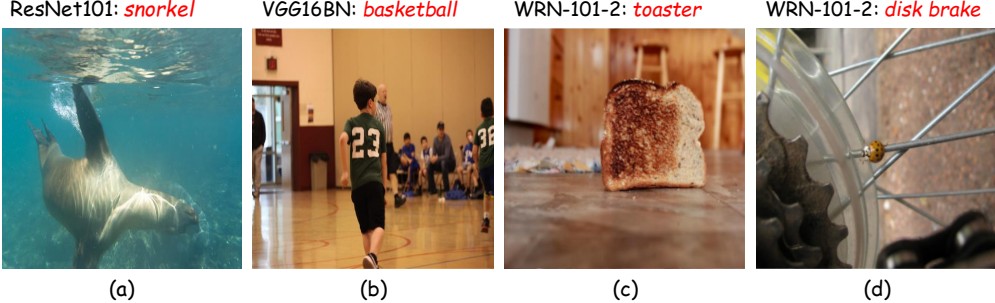

Figure 8: Examples of relation inference. **(a)**: Snorkel is correlated to underwater environment. **(b)**: Basketball is correlated to basketball court. **(c)**: Toaster is correlated to toasted bread. **(d)**: Disk brake is correlated to freewheel and spokes. Similar with how humans recognize objects, it would be reasonable for DNN-based classifiers to make predictions by inferring useful information from object relationships, only when their prediction confidence is low. However, this is not the case in our experiments, which show that classifiers may make high-confidence predictions by leveraging object relations without really "seeing" the predicted object.

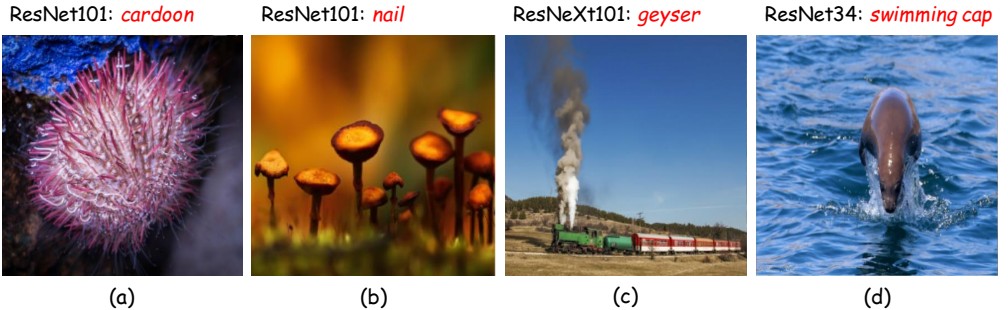

Figure 9: Examples of network bias to low-level visual features, such as color, shape and texture, while overlooking conflicting semantic cues. An ideal classifier is expected to utilize both low-level (appearance) and high-level (semantic) features when making predictions. ResNeXt101-32×4 is abbreviated to ResNeXt101 for neat presentation.

