# OpenReview forum: "I Am Going MAD: Maximum Discrepancy Competition for Comparing Classifiers Adaptively"
_ICLR.cc/2020/Conference — Accept (Poster)_

### Official Review · AnonReviewer2 · 2019-10-23
**Official Blind Review #2**

**Rating:** 8

**Review:**

The paper proposed a novel image classifier comparison approach that went beyond one fixed testing set for all. Instead, for a pair of classifiers to be compared, it advocated to sample their "most disagreed" test set from a large corpus of unlabeled images. The level of disagreement was measured by a semantic-aware distance derived from WordNet ontology. Because of the efficacy of such "worst-case" comparison, the needed set size is very small and thus minimizes the human annotation workload.

The proposed MAD competition distinguishes classifiers by finding their respective counterexamples. It is therefore an "error spotting" mechanism, rather than a drop-in replacement of standard test accuracy. I feel the approach to implicitly assume that the classifiers to be compared are already "reasonably accurate"; since if not, both classifiers might be easily falsified by certain trivial examples, making the "disagreed examples" not as meaningful. If that is true, I would suggest the authors to make this hidden assumption clearer in the paper

The idea shows clear liaison to the "differential testing" concept in software engineering besides the cited work of perceptual quality assessment. The idea has a cross-disciplinary nature and is fairly interesting to me. I can see the paper to be of interest to a quite broad audience and can motivate many subsequent works.

One minor comment: for images in "Case III", the authors considered them "contribute little to performance comparison between the two classifiers" and therefore did not source labels for them. However, since the authors adopted an affinity-aware distance, two incorrect predictions can still be compared based on their semantic tree distances to the true class.


**Experience Assessment:**

I have read many papers in this area.

**Review Assessment: Checking Correctness Of Derivations And Theory:**

I carefully checked the derivations and theory.

**Review Assessment: Checking Correctness Of Experiments:**

I carefully checked the experiments.

**Review Assessment: Thoroughness In Paper Reading:**

I read the paper thoroughly.

---

> ### Author Response · Authors · 2019-11-06
> **Response to Reviewer #2**
>
> 1. I feel the approach to implicitly assume that the classifiers to be compared are already "reasonably accurate"; since if not, both classifiers might be easily falsified by certain trivial examples, making the "disagreed examples" not as meaningful. If that is true, I would suggest the authors to make this hidden assumption clearer in the paper
>
> Response: Thanks for the constructive suggestion. We agree with the reviewer and will make this assumption explicit in the revised manuscript.
>
> 2. The idea shows clear liaison to the "differential testing" concept in software engineering besides the cited work of perceptual quality assessment. The idea has a cross-disciplinary nature and is fairly interesting to me. I can see the paper to be of interest to a quite broad audience and can motivate many subsequent works.
>
> Response: Thanks for recognizing the strengths of the paper. We will add the appropriate references regarding the "differential testing" concept in software engineering.
>
> 3. One minor comment: for images in "Case III", the authors considered them "contribute little to performance comparison between the two classifiers" and therefore did not source labels for them. However, since the authors adopted an affinity-aware distance, two incorrect predictions can still be compared based on their semantic tree distances to the true class.
>
> Response: Thanks for pointing it out. We agree with the reviewer that images falling into Case III can be used to distinguish the associated two classifiers using the proposed semantic tree distance. We will revise the writing to make it more rigorous. In our current subjective assessment environment, we choose to stop labeling images in Case III because it is difficult for humans to select one among 200 classes, especially when they are unfamiliar with the class ontology.

---

> > ### Comment · AnonReviewer2 · 2019-11-10
> > **Thank you**
> >
> > Thank the authors for the fairly prompt feedbacks. The paper’s clarity would definitely benefit from incorporating those responses into the final version.
> >
> > After checking the paper and reviews, I am convinced by this new interesting MAD angle. It could potentially become one big step beyond current accuracy-only evaluations for image classifiers. I vote for accepting this paper.

---

### Official Review · AnonReviewer1 · 2019-10-24
**Official Blind Review #1**

**Rating:** 3

**Review:**

This paper points out that the traditional way of model selection is flawed due to that the validation/test set is often small. The authors also attribute the existence of adversarial examples to the small validation/test set, which I agree to some degree. Hence, the authors proposed an alternative approach to comparing different classification models by the notion of inter-model discrepancy.

The main idea is reasonable, but it requires that the models to compare all perform reasonably well. Otherwise, some poorly performed models could lead to near-random or adversarial inter-model discrepancies, failing the proposed approach.

Another potential issue is that the proposed approach cannot handle training set bias. If all models are biased in similar ways (e.g., toward a particular class or domain), they will not reveal informative discrepancies for the images over which they all make similar mistakes.

Another question which is not answered in the paper is the number $k$ of images to select for each pair of classifiers. Is this number task-dependent? Is it related to the number of classes? What is a general guideline for one to choose this number $k$ given a new application scenario?

The unlabeled set is not "unlabeled" in essence. If my understanding was correct, it cannot contain open-set images which do not belong to any of the classes of interest. It is also nontrivial to control that the images contain only one salient object per image.

Hence, while I agree with the authors that existing approaches to comparing deep neural network classifiers could be improved, I think the proposed solution is not a good alternative yet.



**Experience Assessment:**

I have published in this field for several years.

**Review Assessment: Checking Correctness Of Derivations And Theory:**

I assessed the sensibility of the derivations and theory.

**Review Assessment: Checking Correctness Of Experiments:**

I assessed the sensibility of the experiments.

**Review Assessment: Thoroughness In Paper Reading:**

I read the paper thoroughly.

---

> ### Author Response · Authors · 2019-11-06
> **Response to Reviewer #1**
>
> Regarding comment 1: Thanks for recognizing the merit of our idea. As also mentioned by Reviewer #2, the proposed MAD implicitly assumes that classifiers in the competition are reasonably accurate. Otherwise, the selected counterexamples may be less meaningful. We will make this assumption explicit in the revised manuscript. From this perspective (and other reasons mentioned in the discussion section), MAD should be viewed as complementary to, rather than a replacement for, the conventional accuracy comparison for image classification. When two classifiers perform at a reasonable level and achieve very close accuracy numbers (e.g., VGG16BN and ResNet34 on ImageNet validation set), MAD provides the most efficient way of differentiating the two models by maximizing their discrepancies over a large-scale image set.
> We want to emphasize that MAD is especially useful on image classification tasks where most cutting-edge classifiers achieve very close performance. In these situations, the MAD competition ranking, which is obtained by evaluating on corner examples searched from web-scale unlabeled dataset, is more convincing than something like 1% accuracy advantage on the validation set.
> For problem domains where there are few sufficiently accurate models, we may still apply the underlying principle behind MAD to create adaptive test sets such that the strengths and weaknesses of the models are most easily revealed. In those scenarios, we conjecture that we need increase k to a reasonably larger number, thus at the cost of efficiency.
>
> Regarding comment 2: Thanks for the comment. As long as two (or multiple) models differ (even in slightly different ways), MAD provides the highly efficient way of spotting such differences by exploring a large-scale unlabelled dataset. However, these differences are less likely to be revealed using a fixed and small test set (i.e., they will probably have the same accuracy numbers as models to be compared are very similar and are biased in similar ways). For the extreme case that two models are exactly the same (i.e, they are biased in identical ways and make identical prediction errors), both MAD and traditional accuracy-based methods will draw the same conclusion - the two models have the same performance. Accuracy-based evaluation methods arrive at this conclusion by comparing model predictions with ground truth labels and outputting the same accuracy numbers. In contrast, MAD arrives at the same conclusion without any human labeling since the set S for subjective testing is empty. So in this extreme case, both MAD and accuracy fail to compare those two models.
> In summary, to reliably compare the relative performance of computational models, all evaluation methodologies (including MAD) rely on the assumption that the models to be compared should be diverse to a certain extent, and the proposed MAD makes this assumption more explicit. In fact, MAD makes the best use of model discrepancies (even if models are biased in very similar but not identical ways) to rank the model performance.
> As a matter of fact, based on our experiments, we find that state-of-the-art ImageNet classifiers do have their own biases. (See figure 8 in the appendix.)
>
> Regarding comment 3: Thanks for the excellent question. We believe that the parameter k is task-dependent. For problem domains where there are reasonably accurate models (e.g., imageNet classification in our example), we may obtain a stable ranking with a relative small k (e.g., k=15 in the imageNet classification example). For problem domains where there are no good models, we may increase k to the limit of human labeling budget in order to obtain reasonable performance comparison.
>
> Regarding comment 4: Thanks for the comment. In our current setting, we restrict the dataset D to the domain of interest that contain natural images of mainly 200 classes. However, as the construction of D is noisy and coarse, D contains plenty of open-set images, which do not belong to any class of interest. Since we do not perform any manually data screening at this stage, some of the open-set images may even be selected to construct the dataset S. This means that although the selected open-set image is out of the domain of interest, the associated two classifiers make different, high-confident (with threshold set to 0.8), but incorrect predictions (Case III). As a result, we consider it as a strong counterexample of the two classifiers. Note that this situation rarely happens, at least in our experiments because the competing classifiers tend to give open-set images low-confidence, and therefore are automatically filtered out.
> We agree with the reviewer that selecting images that contain only one salient object requires a lot of human effort. So we did not eliminate Case I. It turns out that keeping Case I does not seem to affect the comparison and analysis of competing models.

---

### Official Review · AnonReviewer3 · 2019-11-11
**Official Blind Review #3**

**Rating:** 3

**Review:**

This paper proposes a new method to compare existing classifiers, which does not use fixed test set and adaptively sample it from an arbitrarily large corpus of unlabeled images. The main idea seems similar to adopting active learning for the test set selection.

One of the main advantage is that it can select a sample set from an arbitrarily large unlabeled images. However, to compare different classifiers, the proposed algorithm still needs humans to annotate the selected dataset, which is very expensive compared with traditional methods.

Since this paper select the top-k images in D, if k is large the annotating for S will be very tedious, however if k is relatively small the method seems very sensitive to selected examples, which will make the comparison not totally convincing.

The authors invite five volunteer graduate students to annotate the selected example. However, for many categories, it’s nor easy for normal people to distinguish. So the experiments in this paper is also not convincing.


**Experience Assessment:**

I have published in this field for several years.

**Review Assessment: Checking Correctness Of Derivations And Theory:**

I carefully checked the derivations and theory.

**Review Assessment: Checking Correctness Of Experiments:**

I carefully checked the experiments.

**Review Assessment: Thoroughness In Paper Reading:**

I read the paper thoroughly.

---

> ### Author Response · Authors · 2019-11-12
> **Response to Reviewer #3 (general reply & question 1)**
>
> With all our due respect to Reviewer #3’s valuable time and effort in reviewing our manuscript, we must admit that we are a bit upset by this last late review, due to the apparent lack of understanding before placing comments, and several factual errors that make the current comments at least poorly grounded.
>
> We understand that the idea of “model falsification as model comparison” might not be trivial to understand for people primarily from practical deep learning backgrounds. The idea is deeply rooted in a successful series of studies from image perceptual assessment research: a basic introduction can be found in (Wang & Simoncelli (2008)). We notice that Reviewer #2 also kindly points out another interdisciplinary foundation of MAD in software differential testing.
>
> We hope Reviewer #3 can carefully read the below explanation, and reconsider the rating to a more serious and appropriate one.
>
> Q1: One of the main advantages is that it can select a sample set from an arbitrarily large unlabeled images. However, to compare different classifiers, the proposed algorithm still needs humans to annotate the selected dataset, which is very expensive compared with traditional methods.
>
> Response:
> Our method is very efficient in terms of human annotation budget compared with traditional methods, which is one of the main claims we elaborated in our paper. We are disappointed that this major important point was not well understood. In fact, MAD provides the very first and efficient solution (in the context of image classification) to exploit a large-scale image set under the constraint of the very limited budget for human labeling.
>
> We have noticed that the other two reviewers agree with us and appreciate this point. For example, quote Reviewer #2: “Because of the efficacy of such "worst-case" comparison, the needed set size is very small and thus minimizes the human annotation workload”.
>
> To evaluate the relative performance of two ImageNet classifiers, traditional evaluation methods compute accuracy on a fixed test set. For ImageNet validation set, human annotations for 50,000 images need to be provided. This number is large in terms of human labeling effort, but is extremely small compared to the set of all natural images (the natural image manifold).  As also mentioned by the reviewer, annotation for each image is a 1000-class classification task, which makes the labeling task more difficult compared to a binary classification problem.
>
> In contrast, rather than comparing fixed test sets which are typically small, the proposed MAD adaptively samples a test set from an arbitrarily large corpus of unlabeled images so as to maximize the discrepancies between the classifiers, measured by the distance over WordNet hierarchy. Human labeling is only required on the resulting small and model-dependent image sets, which contains only k=30 images (for each pair of classifiers) on the ImageNet experiment as reported in our paper. Our experiments show that the MAD ranking stabilizes at around k>15 (see figure 5) and successfully tracks the recent progress in image classification . For comparing 11 classifiers, the total labeled images needed are 1,650 (see page 6): it is obviously smaller than 50,000 and leaves much room to compare more classifiers (before it reaches 50, 000).
>
> In conclusion, our method is apparently much more efficient in terms of human annotation budget compared with traditional methods.
>
> In addition, despite the fact that the selected set by MAD is small (as a way of maximizing the efficiency of human labeling), it provides the strongest examples to let classifiers compete with one another. Quote Reviewer #2: “The proposed MAD competition distinguishes classifiers by finding their respective counterexamples. It is therefore an "error spotting" mechanism”. Their respective strengths, weaknesses as well as biases can be most easily revealed (see figures in the appendix), which sheds light on potential ways to improve the classifiers or combine them into a better one. Those gains are way beyond the scope of collecting random image samples.

---

> ### Author Response · Authors · 2019-11-12
> **[Continued] Response to Reviewer #3 (question 2 & question 3)**
>
> Q2: Since this paper select the top-k images in D, if k is large the annotating for S will be very tedious, however if k is relatively small the method seems very sensitive to selected examples, which will make the comparison not totally convincing.
>
> Response:
> We agree with the reviewer that k is a critical parameter in MAD. We want to however draw the reviewer’s attention to the ablation study and figure 5, if they were accidentally missed in the first reading. Based on them, we cannot concur with the statement “if k is relatively small the method seems very sensitive to selected examples”.
>
> When we apply MAD to compare imageNet classifiers, we find that the MAD ranking stabilizes very quickly when around k>15. We would like to also emphasize that despite the small size of labeled images, MAD successfully tracks the steady progress in image classification, as verified by a reasonable Spearman rank-order correlation coefficient (SRCC) of 0.89 between the accuracy rank on ImageNet validation set and the MAD rank on our test set.
> As also pointed out by Review #2, the selected top-k images provide the strongest examples to let classifiers compete with one another. Through this process, their respective strengths, weaknesses as well as biases can be most easily revealed (see figures in the appendix).
>
>
> Q3: The authors invite five volunteer graduate students to annotate the selected example. However, for many categories, it’s nor easy for normal people to distinguish. So the experiments in this paper is also not convincing.
>
> Response:
> As veterans in performing subjective studies, we understand and agree with the reviewer that querying ground truth labels for a 200-class classification problem is difficult. That is exactly why we have carefully designed our subjective experiment.
>
> Given an image x, which is associated with two classifiers f_i and f_j , we pick two binary questions for human annotators: “Does x contain an f_i(x)?” and “Does x contain an f_j (x)?”. For each question, we follow  the original ImageNet instructions and include the definition of f_i(x) (or f_j(x))  with a link to a corresponding Wikipedia page. We also show several example images of f_i(x) (or f_j(x)) sampled from the ImageNet validation set. Moreover, if more than three of our five human annotators find difficulty in labeling x, it is discarded and replaced.  When both answers to the two binary questions are false (corresponding to Case III), we cease to source the ground-truth label of x for reasons mentioned by the reviewer, and treat x as a strong counterexample for both f_i and f_j.
>
> Based on the above, we cannot concur with the judgement “the experiments in this paper is (are) also not convincing”.

---

### Comment · AnonReviewer4 · 2019-11-12
**Comment/Review #4**

NB this was an emergency review although there is now an R3. I hope the perspective from an additional reader of the paper will be helpful for authors and ACs, but I know there is little time left so it is understandable that authors may not be able to respond.

Summary:

This paper takes a look at the prevailing way of evaluating classifiers through the use of a held-out test set, raising the question of whether this is the best way to be comparing models. It highlights drawbacks of the existing approach, and proposes an alternative which captures more precisely how classifiers can differ even if they have the same accuracy overall. The main proposal is to build a test set adaptively in a manner that captures how classifiers disagree, as measured by the wordnet tree.

Experiments are carried out considering a number of popular models trained on ImageNet, with the unlabelled set of potential test images obtained from Flickr. This is used to re-rank the classifiers, with shifts up to -4.

Review:

I am borderline towards accept.

I find that the strength of the work is the analysis of several recent CNN architectures in a way that moves beyond fixed test set accuracy, and evaluates models through the lens of how their predictions differ. In this sense it is a worthwhile contribution, similar in spirit to work like Recht et al. and Hendrycks et al. in seeking to see how over-reliance on benchmarks can be misleading. However as a general framework it leaves something to be desired, being quite ad-hoc (use of wordnet, choice of k, collecting labels). So I see it as more of a one-off experiment designed to evaluate their hypothesis rather than a methodology people can easily start using to move beyond test accuracy alone.

I think it's good for the community to take a second look at evaluation methodologies, and whether we can be missing something by only caring about being top of the leaderboard, and the experiments support the claims that something is being missed by considering test set accuracy alone, finding more subtle differences between popular models, such as VGG16bn (accuracy 73.36) being re-ranked above ResNet101 (despite higher accuracy 77.37).

"A classifier that is more likely to be falsified is considered worse." How robust is this principle? Would it be thrown off by some pathologically bad classifier in the competition? Would it lead to the Galileo of classifiers being condemned for speaking the truth?

Does it actually solve the problems mentioned? Eg adversarial attack, domain differences. Still need to collect the unlabelled images, which are from a given distribution (Flickr images).

To be useful to the community a good replacement or complementary ranking to accuracy should be easy to report for any new model proposed. But the proposed procedure is a bit unwieldy, it requires having some set of competing classifiers, wordnet classes, and labelling new images for any new classifier added (which may need domain experts). For it to work fairly in practice there'd need to be some sort of evaluation server and annotation effort. Maybe for Imagenet it would be worth it but it's not very general. Alternatively, can any of the principles or lessons learned be used to design a better evaluation protocol but without the expense of manual labelling for each new classifier, such as re-weighting a fixed test set, or to generate some benchmark test sets like ImageNet-A.

"careful inspection of the selected images may suggest potential ways to improve" any examples where this was found?

Feedback:

"we believe a much larger test set in the order of millions or even billions must be used" I think this goes against the rest of the paper (ie a small high quality test set is ok) - and even with infinite budget annotating billions of test images from the same distribution would be of little use.

Presentation: best not to rely on red/green colours being visible (colourblindness or mono printer) add some symbol too

---

> ### Author Response · Authors · 2019-11-13
> **Response to Reviewer #4 (general reply & question 1-2)**
>
> Thanks for the timely and very constructive comments, which provide us directions to further polish our work. Point-for-point responses are as follows.
>
>
> Q1: "A classifier that is more likely to be falsified is considered worse." How robust is this principle? Would it be thrown off by some pathologically bad classifier in the competition? Would it lead to the Galileo of classifiers being condemned for speaking the truth?
>
> Response: Thanks for the excellent questions. The principle of MAD is robust, while its level of effectiveness may be application-dependent. For problem domains where we have reasonably accurate computational models (e.g., imageNet classification investigated in the paper), MAD produces a stable global ranking with a relative small adaptive test set (see ablation study and figure 5). For problem domains where there are no sufficiently accurate computational models, we may have to increase the reliability of the MAD results by increasing k to the limit of the human labeling budget (similarly in this case, a larger test set is needed for reliable accuracy-based model comparison too). This point has also been made explicitly by Reviewer #2, is acknowledged by us, and will be added in the revised manuscript: “the proposed MAD implicitly assumes that classifiers in the competition are reasonably accurate. Otherwise, the selected counterexamples may be less meaningful.”
>
> Nevertheless, in either of the above cases, the selected samples of MAD provide strongest examples to let computational models compete with one another, and only one winner can possibly be declared for each pair of models. In view of this, MAD provides an “error spotting" mechanism (quote Reviewer #2) and maximizes the efficiency of human labeling. It is further useful in revealing the respective strengths and weaknesses of the model pairs, before converting to the ranking results.
>
> Finally, MAD is less likely to be thrown off by some pathologically bad classifier in the competition. This is because the selected samples will finally be subject to reliable human labeling. Although pathologically bad classifiers may reach an agreement on labeling some selected samples as the same incorrect classes, the results from human labeling will stand with the Galileo of classifiers, provided by our carefully designed subjective experiment protocols.
>
>
> Q2: Does it actually solve the problems mentioned? Eg adversarial attack, domain differences. Still need to collect the unlabelled images, which are from a given distribution (Flickr images).
>
> Response: Thanks for the comment. While the existence of natural adversarial examples indeed provides us with part of motivation (see our discussion in the last section), MAD is not designed to solve the adversarial attack problem. We apologize for the confusion, and will revise our manuscript to make the relationship between adversarial examples and samples selected by MAD much clearer.
>
> As for domain differences, a significant advantage of MAD is that it can be applied to an arbitrarily large image set D (on the order of billions) with a closer distribution to the underlying distribution of natural images, “for free”. That is because the human annotation workload in MAD is only dependent on two parameters: number of models to compare, and the hyperparameter k (which is rather small when good classifiers exist, e.g., in our case). It will NOT grow w.r.t. the size of D.
>
> MAD tests the generalizability of the competing classifiers to natural image manifold (represented by D) by only labeling a small and adaptive test set (on the order of thousands). While the current results are based on Flickr crawled data, it is almost free to collect many unlabeled images as we want, from any online image services (such as Flickr, Google, and iNaturalist). It could be MAD’s future extension.
>
> In summary, MAD is a balance between the “large” unlabeled image space to look at and the “small” labeled images eventually needed. It makes a step further towards testing computational models on large-scale databases, compared to accuracy-based evaluation methods on relatively small test sets.

---

> ### Author Response · Authors · 2019-11-13
> **[Continued] Response to Reviewer #4 (question 3 & question 4)**
>
> Q3: To be useful to the community a good replacement or complementary ranking to accuracy should be easy to report for any new model proposed. But the proposed procedure is a bit unwieldy, it requires having some set of competing classifiers, wordnet classes, and labelling new images for any new classifier added (which may need domain experts). For it to work fairly in practice there'd need to be some sort of evaluation server and annotation effort. Maybe for Imagenet it would be worth it but it's not very general. Alternatively, can any of the principles or lessons learned be used to design a better evaluation protocol but without the expense of manual labelling for each new classifier, such as re-weighting a fixed test set, or to generate some benchmark test sets like ImageNet-A.
>
> Response: Thanks for the great comments. We agree with the reviewer that MAD relies on WordNet hierarchy. For problem domains, where no hierarchical structures like WordNet exists, we can use softmax KL divergence as the distance metric, although it has less semantic awareness. Besides, we wish to point out that in many scientific domains, the prior knowledge structures similarly to WordNet do widely exist.  For example, many biomedical imaging ontologies have been under development for years and gain popularity, e.g., in histopathological image understanding. For those domains, MAD actually presents another means of incorporating domain knowledge into data-driven model evaluation. We may also define a domain-specific cost measure that gives different weights to different classes. We will explore this in our future work.
>
> We agree that labelling new images when any new classifier is added may take effort and require domain experts. For MAD, it is straightforward and relatively cost-effective to add the m + 1-th classifier into the current MAD competition. No change is necessary for the sampled S and the associated subjective testing. The only additional work is to select a total of mk new images from D for human labeling (see Algorithm 2). In the near future, we plan to organize a challenge to facilitate the use of MAD in machine learning and computer vision, with the evaluation server and the annotation tool well provided. We welcome your valuable inputs, comments and suggestions on how we could make MAD more broadly useful for the research community.
>
> We also agree that MAD needs some set of competing classifiers in order to rank another classifier, which is the case in most application scenarios. Moreover, MAD mainly aims at relatively comparing and ranking classifiers instead of giving an absolute value indicating the performance of a certain classifier.
>
> Thanks for your great suggestion on re-weighting a fixed test set, or to generate some benchmark test sets like ImageNet-A. It is definitely an interesting direction that is worth exploring, given that we have a well-defined semantic tree distance in Eq. (3).
>
>
> Q4: "careful inspection of the selected images may suggest potential ways to improve" any examples where this was found?
>
> Response: Yes, please kindly check Figures 8, 9, 10  (on page 12 in Appendix) and Figure 3 (on page 7) in our paper, each of which reveals the relative pros and cons of the two models compared. On top of the spotted problems or model biases, one straightforward application is to choose a model that can deliver your “preferred behavior” regarding a specific application. Additionally,  model ensembles could be formed to alleviate the bias of each individual model. Taking Figure 8 just for a concrete example: if some applications emphasize picking up foreground and smaller objects, then WSL-ResNeXt101-32×48 could be a better option than EfficientNet-B7. Further, the ensemble of WSL-ResNeXt101-32×48 and EfficientNet-B7 may yield a more comprehensive capability of accurately recognizing in varying scales. We would leave more application-specific exploration as future work.

---

> ### Author Response · Authors · 2019-11-13
> **[Continued] Response to Reviewer #4 (question 5 & question 6)**
>
> Q5: "we believe a much larger test set in the order of millions or even billions must be used" I think this goes against the rest of the paper (ie a small high quality test set is ok) - and even with infinite budget annotating billions of test images from the same distribution would be of little use.
>
> Response: Thanks for the comment. We can see where your confusion was: the “millions or even billions” order refers to the size of unlabeled image space D, not to the desired amounts of annotations.
>
> In fact, as we respond to Comment #2, MAD is a balance between the “large” unlabeled image space to look at and the “small” labeled images eventually needed. A significant advantage of MAD is that it can be applied to an arbitrarily large image set D (on the order of billions) with a closer distribution to the underlying distribution of natural images, “for free”. That is because the human annotation workload in MAD is only dependent on two parameters: number of models to compare, and the hyperparameter k (which is rather small when good classifiers exist, e.g., in our case). It will NOT grow w.r.t. the size of D.
>
> We apologize for having caused your confusion and hope the above explanation clarifies it. We will definitely revise this statement in the paper.
>
>
> Q6: best not to rely on red/green colours being visible (colourblindness or mono printer) add some symbol too.
>
> Response: Thanks for the suggestion. We will make sure to add different symbols to indicate correct and incorrect predictions, respectively.

---

### Decision · Program_Chairs · 2019-12-19

**Decision:**

Accept (Poster)

**Comment:**

This paper proposes a new way of comparing classifiers, which does not use fixed test set and adaptively sample it from an arbitrarily large corpus of unlabeled images, i.e. replacing the conventional test-set-based evaluation methods with a more flexible mechanism. The main proposal is to build a test set adaptively in a manner that captures how classifiers disagree, as measured by the wordnet tree. As noted by R2, this work has the potential to be of interest to a broad audience and can motivate many subsequent works.

While the reviewers acknowledged the importance of this work, they raised several concerns: (1) the proposed approach is immature to be considered for benchmarking yet (R1,R4), (2) selecting k and studying its influence on the performance ( R1, R3, R4), (3) the proposed approach requires data annotation which might not be straightforward -- (R3, R4).  The authors provided a detailed rebuttal addressing the reviewer concerns.

There is reviewer disagreement on this paper. The comments from R3 were valuable for the discussion, but at the same time too brief to be adequately addressed by the authors. The comments from emergency reviewer were helpful in making the decision. AC decided to recommend acceptance of the paper seeing its valuable contributions towards re-thinking the evaluation of current SOTA models.